# *In situ* study on atomic mechanism of melting and freezing of single bismuth nanoparticles

Yingxuan Li[1,2], Ling Zang[3], Daniel L. Jacobs[3], Jie Zhao[1,2], Xiu Yue[1,2] & Chuanyi Wang[1,2]

Experimental study of the atomic mechanism in melting and freezing processes remains a formidable challenge. We report herein on a unique material system that allows for *in situ* growth of bismuth nanoparticles from the precursor compound $SrBi_2Ta_2O_9$ under an electron beam within a high-resolution transmission electron microscope (HRTEM). Simultaneously, the melting and freezing processes within the nanoparticles are triggered and imaged in real time by the HRTEM. The images show atomic-scale evidence for point defect induced melting, and a freezing mechanism mediated by crystallization of an intermediate ordered liquid. During the melting and freezing, the formation of nucleation precursors, nucleation and growth, and the relaxation of the system, are directly observed. Based on these observations, an interaction–relaxation model is developed towards understanding the microscopic mechanism of the phase transitions, highlighting the importance of cooperative multiscale processes.

[1] Laboratory of Environmental Sciences and Technology, Xinjiang Technical Institute of Physics & Chemistry, Chinese Academy of Sciences, Urumqi 830011, China. [2] Key Laboratory of Functional Materials and Devices for Special Environments, Chinese Academy of Sciences, Urumqi 830011, China. [3] Nano Institute of Utah and Department of Materials Science and Engineering, University of Utah, Salt Lake City, Utah 84112, USA. Correspondence and requests for materials should be addressed to Y.L. (email: yxli@ms.xjb.ac.cn) or to C.W. (email: cywang@ms.xjb.ac.cn).

Melting and freezing, as first-order phase transitions between liquid and solid, have long been the research subjects across many disciplines including physics[1], chemistry[2], biology[3] and material science[4]. Traditional understanding of the nucleation and growth process in melting and freezing is based on classical nucleation theory (CNT), which assumes that a particle will nucleate at a site and grow outwards via atomic migration and addition. However, a recent study has recognized that the CNT has some limitations in describing the details of the nucleation process in phase transitions due to its simplifying assumptions[5]. A detailed understanding of the atomic mechanisms controlling the reversible transitions beyond CNT has broad and increasing interests for fundamental science and applications as it may provide the necessary insights into precise control of the two processes. Progress in understanding the details of the phase transitions has led to development of alternative mechanisms to the CNT, such as two-step nucleation mechanism in crystallization and solid–solid phase transitions[6,7] and a defects/dislocations induced melting mechanism[5,8]. Recently, Ostwald's step rule[9], which states that phase transitions proceed through metastable states with the free energy closest to the parent phase, was confirmed to play a key role during the melting of superheated colloidal crystals[10], and crystallization of a metal phosphate[11]. Clearly, there is a growing consensus that the CNT is lacking important metastable transition states, but comprehensive theoretical and experimental evidence of these states is still lacking.

Because melting and freezing are rare barrier-crossing events[5], the intricate dynamics controlling the microscopic mechanism of the two processes is difficult to predict by theoretical modelling[12]. For example, the discrepancies of crystallization rates between the computational predictions and experimental results can be up to several orders of magnitude[13,14]. Such large discrepancies highlight the need for more precise and detailed experimental results to justify the theoretical models and to further our understanding of the melting and freezing processes. However, experimentally studying these microscopic mechanisms remains a significant technical challenge as it demands an experimental system that not only enables manipulation between different phases, but also allows for *in situ* imaging of the atomic structural change during phase transition. Furthermore, experimental designs to study the corresponding dynamics are significantly limited by the very fast timescales in which these transitions occur[10]. Therefore, many key aspects of the phase transition, such as nucleation events and metastable intermediates, are rarely observed. To overcome these limitations, colloidal systems, in which the transition dynamics is fairly slow and the particles are comparatively large, have been used to investigate the primary process of phase transitions[7,8,10–12,15–17]. However, relatively little is known about the atomic origins of melting and freezing in real systems, especially at the very early and final stages of the transitions. For one, compared with the liquid freezing process, it is more difficult to study the precise microscopic mechanism in a melting event because, unlike supercooling a liquid, superheating a solid crystal leads to heterogeneous nucleation of melting from surfaces or grain boundaries[8,10,15]. Moreover, while most reports limit the study of the nucleation or growth process to a local event[7,8,10,15,16], the nonlocal behaviour also contributes to the melting process of a solid as found in a recent simulation[5]. Thus, atomic information on the unabridged pathway of both transitions from early nucleation stage to the final transition of the entire system is of fundamental importance to understanding the mechanisms of the reversible phase transitions.

Experimentally, extracting information about an individual event is critical to understanding actual phase transition mechanism, and this requires close observation of any local variation of the phase properties to observe the local structural inhomogeneity. In bulk or ensemble systems, the phase transition proceeds through several nucleation events, and the kinetic understanding is developed around the statistical averaging of these events[18,19]. Thus, many key aspects of the phase transition may be buried in these parallel events and cannot be observed directly[19,20]. For a nanoparticle, a single nucleation event typically controls the freezing/melting of the particle. Therefore, an individual nanoparticle would be an excellent candidate for studying the nucleation and growth mechanisms in phase transitions[19,20]. To our best knowledge, although the phase transitions of nanoparticles have been widely studied[21–27], the complete information and direct observation of melting and freezing dynamics of a single nanoparticle from initial stage are rarely reported. Furthermore, in previous reports, melting and freezing events are usually studied separately[1,5,6,8,10,15,16]. If this reversible process could be simultaneously investigated in the same system by precisely controlling the temperature, it will be beneficial to understanding the two closely related processes systematically. The challenge for achieving this arises from the fact that the reversible freezing/melting process of a single nanoparticle is more difficult to reach and maintain because of differences in free energy barriers for the forward/backward phase transitions and densities between the two phases[13]. However, work from Wagner et al.[28] has shown that bismuth (Bi) nanoparticles can undergo a controllable and reversible freezing/melting process under electron-beam irradiation in a high-resolution transmission electron microscope (HRTEM) by controlling the incident current and thereby nanoparticle temperature.

In this work, we use Bi nanoparticles as a model system to probe the metastable states of the reversible phase transitions by *in situ* HRTEM. The study provides clear atomic-scale evidence for point defect induced melting, and a freezing mechanism mediated by crystallization of an intermediate ordered liquid. Based on the direct observations, we prove that the two opposite processes of melting and freezing are controlled by the interaction–relaxation model, which might provide guidance for future theoretical investigation and for better control of the dynamics of the phase transition.

## Results

**Initial freezing of Bi nanodroplets**. Liquid Bi nanoparticles (nanodroplets) were grown from a precursor compound ($SrBi_2Ta_2O_9$) under electron-beam irradiation in a HRTEM based on our previous work[29]. The formation of elemental Bi was confirmed by energy dispersive spectroscopy (EDS) in Supplementary Fig. 1. The nanodroplets exhibit a slip-stick movement characteristic of liquid as shown in Supplementary Movie 1, which leads to growth of nanodroplet via coalescence (Supplementary Movie 2). Under constant incident electron irradiation, the nanodroplets maintain a liquid phase, due to the low melting temperature of Bi nanoparticles[29], until a critical point is reached. At this point, the particle undergoes an initial liquid–solid phase transition, after which continuous and stable electron irradiation initiated a unique state of perpetual melting and freezing without the need to tune the electron-beam current. This enables continuous imaging of the reversible phase transitions and offers unprecedented access to study the atomic mechanisms of the metastable transition states leading to the reversible melting/freezing processes.

A typical real-time video before the initial freezing of a Bi nanodroplet is shown in Supplementary Movie 3, which was recorded between 1,018 and 1,042 s after electron-beam

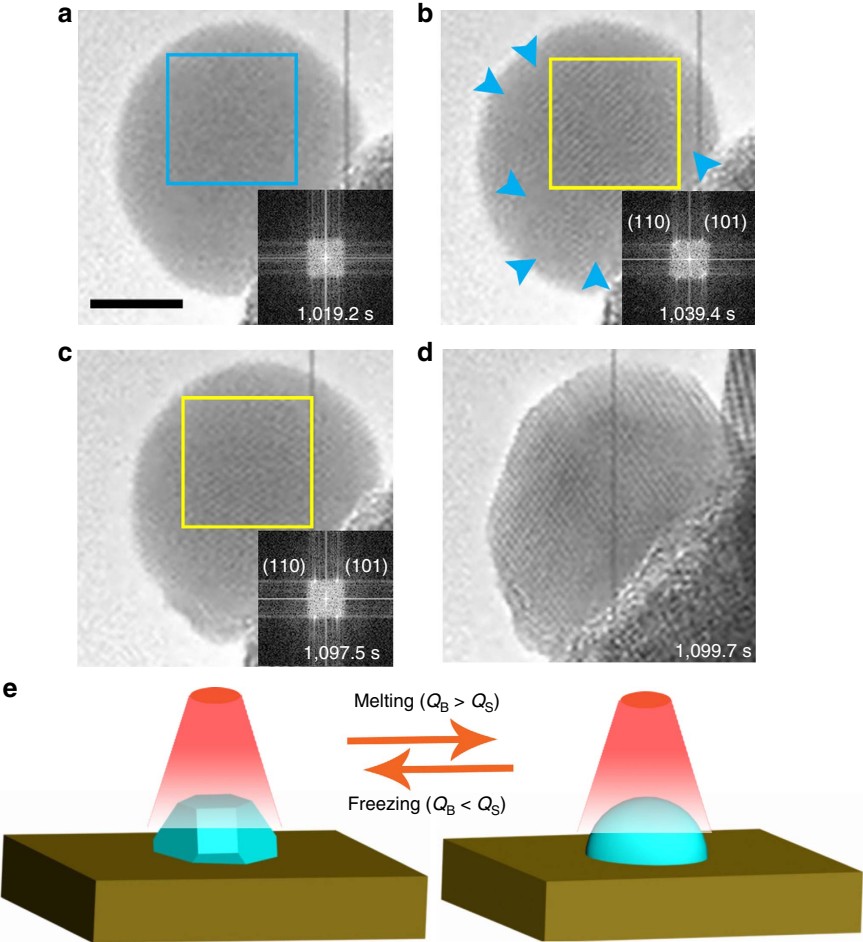

**Figure 1 | The metastable state and crystallization of a Bi nanodroplet.** (**a**,**b**) Sequential snapshots of HRTEM imaging showing the metastable state of the Bi nanodroplet. The blue arrows in **b** indicate the domains still retained in liquid phase. (**c**,**d**) Sequential snapshots of HRTEM imaging showing fast crystallization of the nanodroplet after the induction period. The insets show the corresponding FFT patterns of the nanoparticle. (**e**) Schematic illustration of the reversible freezing/melting of the Bi nanoparticle. The scale bar in **a** is 5 nm, which applies to **a**–**d**. The time labels correspond to when the video snaps were taken.

irradiation. The nanodroplet under investigation is the left-most particle found in Supplementary Movie 3, and individual frames of the nanoparticle selected from the video are shown in Fig. 1a,b. At the time of 1,018 s when this video starts, the sample was already under the electron-beam irradiation for ~17 min, and thereby it can be assumed that the Bi nanodroplet and $SrBi_2Ta_2O_9$ substrate are in an equilibrium state. This is consistent with the observation that the size of the nanodroplets remained unchanging in the process, that is, no further growth of the droplets was observed. As shown in Fig. 1a, the nanodroplet shows a uniform contrast, suggesting that it remains in a liquid phase[30], which is supported by the fast Fourier transform (FFT) pattern shown in the inset of Fig. 1a.

Careful examination of Supplementary Movie 3 reveals that the liquid Bi nanodroplet actually exists in a metastable liquid state in which frequent formation and dissolution of crystallized clusters were observed inside the nanodroplet. These crystalline phases within the liquid nanodroplet can be seen in Fig. 1b. The lattice spacing of temporally crystallized clusters in the nanodroplet is 0.36 nm, which is consistent with Bi (101) planes, indicating that the atom arrangement in the embryo is identical to that in a bulk crystal. Interestingly, the inset FFT pattern in Fig. 1b shows that these crystallized clusters share the same crystal direction, implying that the system has long-range ordering interactions. To support the long-range

orientational correlation in the metastable nanoparticle, sequential snapshots of HRTEM imaging were sampled every second during the metastable state over a period of 17 s (Supplementary Fig. 2). In this period, the nanoparticle was in a continual flux of crystallite formation and dissolution. As seen in the associated FFT patterns, there is a single orientation throughout the entire time signifying long-range ordering across the particle consistent across a relatively long time-frame. Despite the evidence of crystalline phases in the droplet, strong surface oscillation of the nanoparticle shown in Supplementary Movie 3 is characteristic of an overall liquid phase.

**Origin of the reversible phase transitions of Bi nanoparticles.** Plenty of studies have found that the electron-beam irradiation in TEM can heat samples due to the inelastic scattering of incoming electrons[19,28,31,32]. While heating with an electron-beam irradiation is physically and mechanistically different than thermal heating, the use of the electron beam as a heater has previously been reported to induce a reversible phase transitions of Bi nanoparticles[28], and a structural transformation of a $Cu_2S$ nanorod[19]. Therefore, in the present study, the electron-beam induced phase transitions of Bi nanoparticles while imaging by TEM is proposed to be similar to a simple heating effect. In

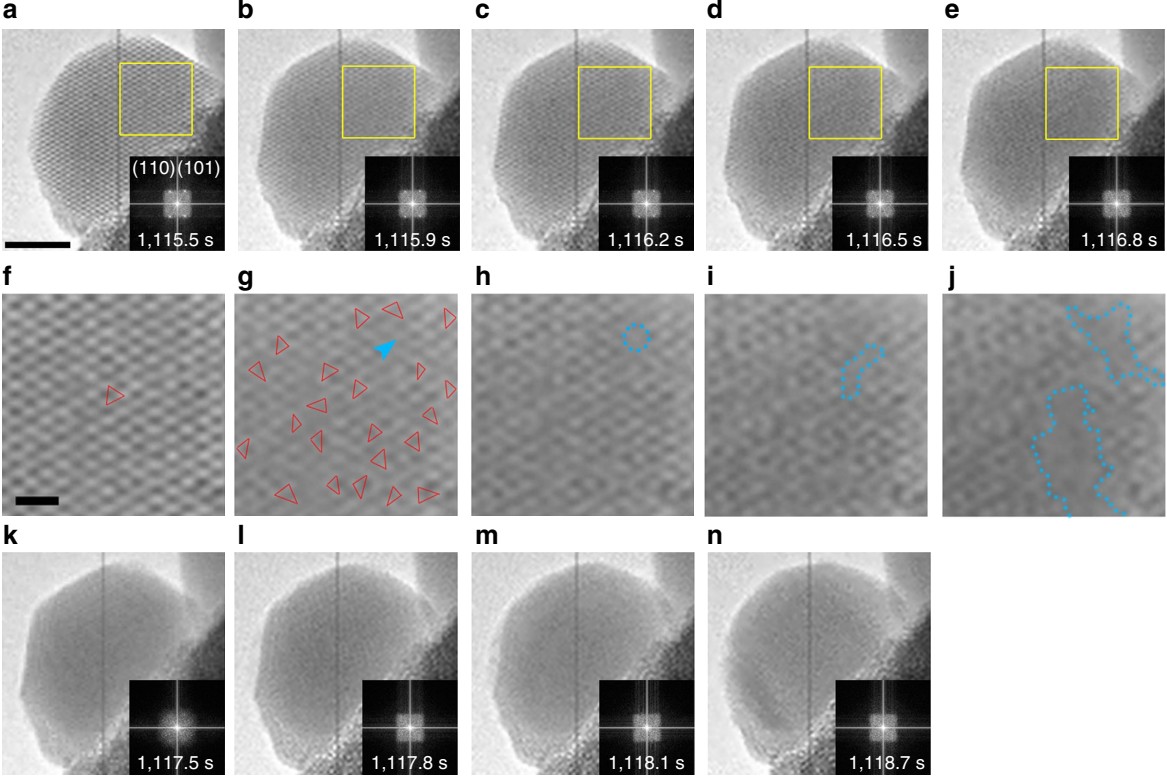

**Figure 2 | Point defect induced melting of Bi nanocrystal. (a–e,k–n)** Sequential snapshots of HRTEM imaging showing the microscopic structural details of melting process. **(f–j)** The enlarged images corresponding to the regions as marked by the yellow squares in **a–e**. The lattice fringes in **f,g** are highlighted by red lines. The dashed blue circle in **h** indicates a vacancy. The blue arrow in **g** indicates the atomic column before transforming to the vacancy in **h**. The area marked in dashed blue line in **i** indicates the gap formed by the coalescence of the defects. The areas marked in dashed blue lines in **j** indicate the domains in liquid-like structure with an irregular shape. The insets of **a–e** and **k–n** show the corresponding FFT patterns of the nanoparticle. The scale bar in **a** is 5 nm, which applies to **a–e** and **k–n**. The scale bar in **f** is 1 nm, which applies to **f–j**.

the equilibrium state, the heat flowing to the Bi nanodroplet induced by the electron beam ($Q_B$) balances the heat radiation from Bi nanodroplet to the surroundings ($Q_S$), which suggests that the metastable liquid Bi nanodroplet is at a supercooled temperature (that is, an equilibrium temperature that is sustained by the electron beam, but lower than the melting point of the nanodroplet determined by the size). The metastable state as shown in Supplementary Movie 3 persisted for a relatively long time (~80 s) until it reached a critical point at 1097.5 s (Fig. 1c). At this critical point, the nanodroplet underwent a sudden shape change from spherical to a faceted morphology (Fig. 1d), which is accompanied with a rapid crystallization process.

After the initial crystallization in Fig. 1, the whole-Bi nanoparticle remains in a continual reversible freezing/melting state until the end of the video recorded (at 1,344.0 s). The liquid phase of the nanodroplet in the reversible process was also confirmed by the HRTEM image after the end of the video (Supplementary Fig. 3). The mechanism that induces such a reversible transition is ascribed to the change of the thermal conductivity of Bi before and after melting, which is illustrated in Fig. 1e. As we know, solid Bi is a semimetal with poor thermal conductivity, while liquid Bi is metallic with high thermal conductivity[22,28]. Under electron-beam irradiation, the freshly formed Bi nanocrystal in Fig. 1d was superheated because of the poor thermal conduction of solid Bi ($Q_B > Q_S$), thus melting of the solid Bi occurs as shown in Fig. 2. After melting, the liquid sample began to cool by conductive heat loss to the surroundings because liquid Bi is a metal with a higher thermal conductivity. The thermal contact areas between

the Bi liquid and the support before and after the first crystallization of the nanoparticle at 1,099.7 s are 55.4 nm² and 117.9 nm², respectively (Supplementary Fig. 4), which results in more heat loss from Bi liquid to the support compared with the metastable liquid in Fig. 1, and thus leads to the deep supercool and recrystallization of the nanodroplet as shown in Fig. 3. With this slow and self-controlled oscillation around the melting temperature, the Bi nanoparticle can remain in the reversible freezing/melting state under constant experimental conditions. As a result, the sequential and discrete atomic structural changes and dynamics leading to the melting and freezing processes can be captured by *in situ* HRTEM imaging. This unconventional phenomenon makes Bi nanoparticle a fascinating candidate for studying the microscopic mechanisms of melting and freezing phenomena near the melting temperature.

To understand the temperature and heat flow properties of this system, the temperature range of melting and heating is analysed. The melting temperature of the Bi nanoparticles is known to be proportional to the reciprocal of the radius of the particles[22], and the radius of the nanoparticle shown in Fig. 1 is determined to be ~8 nm. Differential scanning calorimetry (DSC) was used to determine the temperature range over which this reversible reaction occurs. However to achieve the large density of nanoparticles needed for a DSC signal, the Bi particles were grown on the SrBi₂Ta₂O₉ substrate via a photoreduction method introduced in our previous work[33]. The TEM image (Supplementary Fig. 5) shows that the radius of the nanoparticles formed on the surface of the platelet after 40 h photoreaction is in the range of 6–9 nm, which is

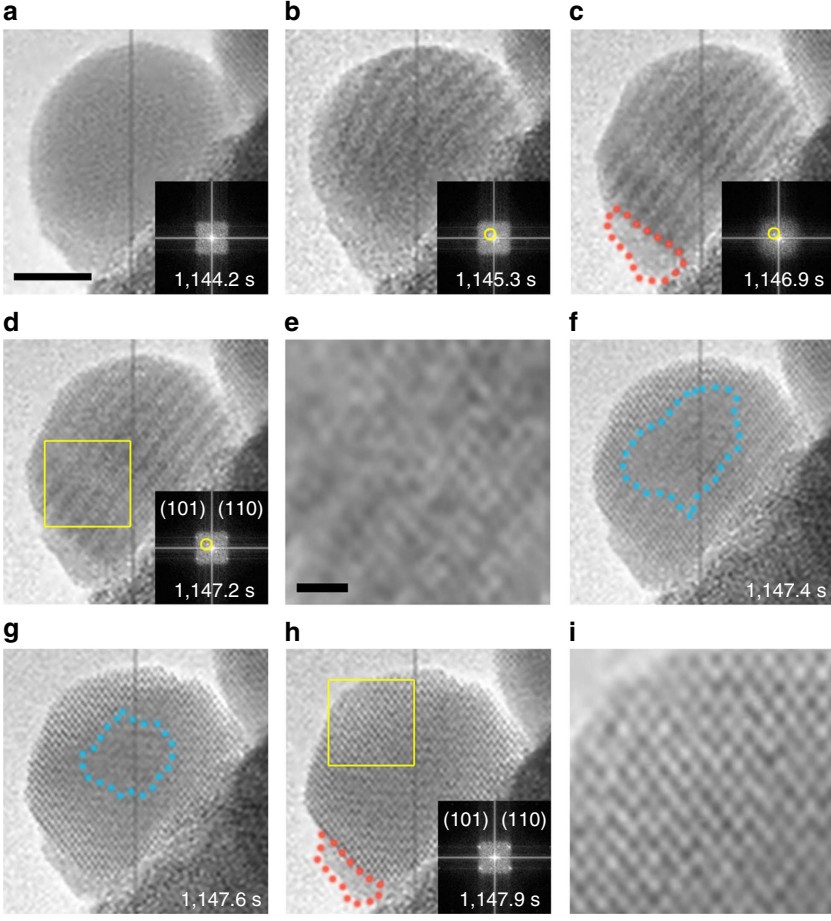

**Figure 3 | Crystallization of a Bi nanodroplet by an intermediate ordered liquid.** (**a–d,f–h**) Sequential snapshots of HRTEM imaging showing a crystallization through transformation of ordered liquid structure. The insets show the corresponding FFT patterns of the nanoparticle. The spots indicated by yellow circles in the FFT patterns in **b–d** reflect the formation of the periodic structure. The areas marked in dashed blue lines in **f,g** indicate the domains in disordered structure. The enlarged images of the regions as marked by the yellow squares in **d,h** are shown in **e,i**, respectively. The part that cannot transform into the well crystallized phase is indicated by the dashed orange line. The scale bar in **a** is 5 nm, which applies to **a–d,f–h**. The scale bar in **e** is 1 nm, which applies to **e,i**.

similar with that in Fig. 1. The heating and cooling DSC curves (Supplementary Fig. 6) indicate that the onset melting and onset freezing temperature of these nanocrystals are 188 °C and 160 °C, respectively. The particles grown in the photoreduction method exhibit a small size variation spanning across the particle size grown under electron-beam radiation. Thus, the phase transition temperature range produced under electron-beam irradiation on the single particle formed under HRTEM should be reasonably close to the measured range of 160–188 °C, assuming the effect of the electron beam can be modelled as a simple heating source as argued above.

**Melting trajectory of Bi nanocrystals.** Following the initial freezing shown in Fig. 1, the melting dynamics was directly observed for the same Bi nanoparticle as shown in Supplementary Movie 4, from which nine snapshots are selected and presented in Fig. 2. As shown in Fig. 2f–j, the enlarged image for each region marked by the yellow square in Fig. 2 a–e appears beneath the corresponding image. Initially, the entire nanoparticle was well crystallized, and possesses almost no defects (Fig. 2a,f). At 1,115.9 s, the particle underwent a fast shape change, but the overall crystal structure remained (comparing Fig. 2a,b). The shape change was a result of collective motion of interior atoms leading to fast atomic displacement at angstrom (Å) scale over the

whole nanoparticle as seen in the variations of lattice fringes in Fig. 2g (outlined with red lines). The sudden lattice displacement can be regarded as an isostructural solid–solid phase transition caused by the competition between the interaction energy and the free-volume entropy[34]. Careful analysis of FFT patterns from Fig. 2b–e,k reveals that the diffraction spots corresponding to the crystal lattices gradually disappear, suggesting increasing disorder occurring throughout the crystal. The observed disordered structure can be regarded as premelting of the nanocrystal[8], which is triggered by the solid–solid phase transition in Fig. 2a,b. The similar solid–solid phase transition has been observed to induce the surface premelting in colloidal crystals[34]. Within the framework of premelting, a point defect induced nucleation in melting process of the nanocrystal was observed as shown in the images in Fig. 2c–n. It is conventionally believed that premelting occurs at the surfaces of the melting solid or locally from the defects within the solid[8,13], however, it is shown in Fig. 2b–e that the disorder occurs uniformly across the crystal and not confined locally to the defect area.

At 1,116.2 s, a vacancy (highlighted by a dashed blue circle in Fig. 2h) is generated from the right part of the disordered nanocrystal. The evolution of the vacancy can be seen in the intensity profiles shown in Supplementary Fig. 7. Formation of the defect is likely a result of the competition between entropy and enthalpy change, and is known to increase the free energy of

a crystal[5]. Following that, the defects start to aggregate into a gap (indicated by dashed blue lines) within the nanocrystal to lower the free energy (Fig. 2i). The gap induces the formation of the crystallite interfaces in the nanocrystal. To minimize the interfacial free energy caused by stress and surface tension, areas near the grain boundary start to melt. Thus, an erupted formation of two liquid nuclei with an irregular shape near the gap (indicated by dashed blue lines in Fig. 2j) was observed, indicating the defects play an important role in the heterogeneous nucleation during the melting of the nanocrystal. The intensity profile of the selected area in Fig. 2e clearly reveals the presence of a liquid phase (Supplementary Fig. 8). As indicated by the FFT patterns, a subsequent rapid melting of the entire nanoparticle occurs between 1,116.8 and 1,117.5 s (Fig. 2e,k). It is worth noting that the nanoparticle still maintains a faceted shape at this time.

In some sense, a TEM image can be considered as a two-dimensional projection of three-dimensional features. As we know, the electrons in TEM can penetrate the samples with a thickness of about 100 nm (ref. 35). For the nanoparticle with a maximum thickness of $\sim 15$ nm in the present work, although we cannot fully discern structures between internal nanoparticle and surface by TEM imaging, the crystallized clusters should be easily detected if they indeed exist regardless of if they are inside or outside the nanoparticle. Moreover, the observation of HRTEM lattice fringe images in Fig. 2a clearly prove that the crystallized clusters could be observed under the current experimental conditions. Therefore, uniform contrast of the faceted nanoparticle and the FFT pattern of the HRTEM image in Fig. 2k indicate that the main body of the nanoparticle is in a liquid state at this time. The formation of a droplet with planar surfaces is extremely rare and somehow counterintuitive considering that liquid materials are generally in isotropic disorder phase[36]. Following that, the nanoparticle immediately starts a gradual transformation from the transient faceted shape to spherical morphology (see details in Fig. 2k–n and Supplementary Movie 4). The liquid nature of the nanoparticle through the faceted and spherical regime was confirmed by FFT patterns in the inserts of Fig. 2k,n.

It is well known that melting should start at the surface of the solid. However, as seen in Fig. 2, no such surface melting or roughening occurs on the facets of the Bi nanocrystal until the entire particle melts. A similar phenomenon has been previously reported in other Bi nanoparticle systems[22], which is attributed to the interactions between the liquid–vapor and solid–liquid interfaces. The interaction force (per unit area) of the nanodroplet can be calculated by[37]

$$F = -2H/l^3 \qquad (1)$$

where $H$ is the Hamaker constant dependent on the dielectric constant of the material, and $l$ is the distance between the interfaces. Bi is an unusual material that has a negative $H$ ($< -1.3 \times 10^{-21}$ J) because solid Bi is a semimetal, while liquid Bi is metallic[22]. This means that the interaction between the solid–liquid interface and the liquid–vapor interface will be attractive, which will induce an energetic barrier to suppress the nucleation of the melt at the surface of the Bi nanoparticle[22,37]. Therefore, a superheat of the Bi nanoparticle is easily achieved, which further enhances it as a good candidate for studying the detailed melting trajectory in atomic scale.

**Freezing trajectory of Bi nanodroplets**. The metastable state of continuous freezing/melting was observed for several cycles as shown in the 24 s period of Supplementary Movie 5, enabling repetitive observations of transition states for both phase transitions. In addition to the premelting transition state described above and in Fig. 2, a novel two-step crystallization process via an intermediate ordered liquid transition state and prefreezing state was also observed in the same Bi nanoparticle as shown in Fig. 3 and Supplementary Movie 5. The two-step crystallization process starts with an isotropic liquid (Fig. 3a) and forms a liquid state with a periodic ordering as imaged by HRTEM (Fig. 3b,c). Generally, the periodic contrast pattern in the HRTEM image (Fig. 3c) reflects the periodic variation of the atom thickness in the nanodroplet. One set of spots in the FFT patterns in the inserts of Fig. 3b–d could be identified based on the d-spacing ($\sim 1.2$ nm) of the periodic structure. The sharp structural change shown in Fig. 3a–c suggests a collective atomic motion in the nanoparticle. The liquid feature of both the isotropic and ordered structures was confirmed by the FFT patterns shown in the insets of Fig. 3a–c. These results to some extent prove the formation of the ordered liquid in Fig. 3c. However, the underlying mechanisms for the formation of the ordered liquid remain unclear. Thermal capillary waves or periodic waves caused by heat flux are proposed as the possible reasons involved therein[38,39], but these conjectures could not be confirmed.

After the formation of ordered liquid, the crystallization of the nanoparticle is divided into two distinct stages. In the first stage, the ordered liquid acts as a template for crystallization of the liquid layers (Fig. 3d), but does not induce a change in the overall shape of the nanodroplet. The formation of ordered liquid lowers the entropy of the nanodroplet, while the enthalpy change in the process can be neglected as both the initial and final states were in liquid phase. Therefore, the formation of ordered liquid at this stage allows for a large driving force (that is, free energy change) to crystallization as shown in Fig. 3d. The inset FFT pattern in Fig. 3d demonstrates that the crystallized layers share the same crystal direction, and the local structures formed have long-range ordering interactions. However, as seen from the enlarged image in Fig. 3e, atomic displacement obviously exists in the whole nanoparticle compared with the well crystallized nanoparticle in Fig. 3i. Therefore, the formation process of the disordered crystal in Fig. 3a–d can be regarded as the prefreezing of nanoparticle similar to the premelting from Fig. 2. In the second stage of this freezing process, the crystallized layers in the prefreezing state interact with each other to develop a collective behaviour of crystallization from the outer surface of the nanoparticle (Fig. 3d–i), which results in the gradual formation of a well faceted nanocrystal. As shown in Fig. 3f–h, the overall change in the shape of the nanoparticle was achieved by the collective motion of the atoms along a transition path. Interestingly, a very small portion of the nanodroplet with a uniform contrast was also found throughout the freezing process along the left bottom edge of the particle in Fig. 3 (marked by dashed orange line). This remnant liquid area also did not undergo the ordered liquid state, suggesting that the ordered liquid shape is a crucial initial step towards crystallization.

**Crystalline orders in melting and freezing trajectories**. Our observations indicate that the melting and freezing of the nanoparticle should be regarded as cooperative multiscale processes, revealing the importance of collective behaviour that is inaccessible in previous experiment. Therefore, finding a suitable parameter that can characterize the overall feature of the transforming system is a crucial step to estimate the phase transitions. It is known that phase transitions between liquid and solid describe a transformation between a disordered and an ordered state. Therefore, as shown in Fig. 4, the phase transitions of the nanoparticle were quantified by calculating

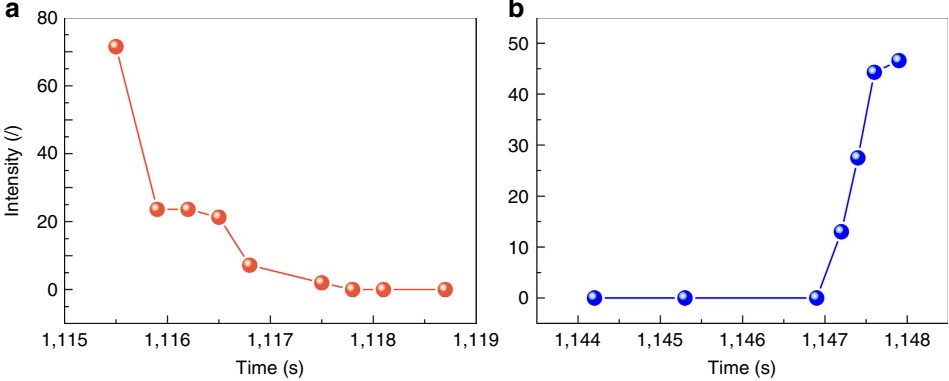

**Figure 4 | Quantifying the phase transitions of the nanoparticle.** The amount of crystalline order was measured from the integrated profiles taken from the 2D FFT images. The 2D FFT images in Figs 2 and 3 were transformed to FFT spectra by the Fit2D program (see details in Supplementary Fig. 9). The distinctive peaks on the integrated profiles were produced by pixel intensities of the spots in the FFT patterns. (**a,b**) The intensity of the left-most peak in the integrated profile, $I$, corresponds to the time labels in Figs 2 and 3, respectively.

the crystalline order of the entire nanoparticle as a function of time. The amount of crystalline order of the nanoparticle can be determined from the FFT patterns of the two-dimensional TEM images. Commercial software (WSxM 5.0 Develop 6.0) was used for image processing according to previous reports[40,41]. The two-dimensional FFT patterns were integrated into one-dimensional data using the Fit2D programme, which is usually used for integrating two-dimensional X-ray diffraction images (see details in Supplementary Fig. 9)[42,43]. As shown in Supplementary Fig. 9, the Fourier spectrum is essentially the sum of the FFT outputs from individual atom arrays. Therefore, in the reversible phase transitions, the crystalline order for a nanoparticle is quantified as a function of time by calculating the peak intensity of the Fourier spectrum, $I$. As shown in Fig. 2l–n, the FFT spots are indistinguishable from the background, and the $I$ values for these images are defined as 0. The evolution of the peak intensity versus irradiation time in the melting process is shown in Fig. 4a (corresponding to the time labels in Fig. 2). As shown in Fig. 4a, a precipitous decrease of the $I$ is observed in the formation process of the premelting state between 1,115.5 and 1,115.9 s. After that, there is no obvious change in the crystalline order before 1,116.5 s. However, the other notable decrease of the crystalline order was observed between 1,116.5 and 1,118.7 s, in which the nucleation and growth processes of the melting occur. Based on the quantitative analysis in Fig. 4a, we can conclude that the change of the crystalline order in the melting process is stepwise. However, as shown in Fig. 4b, the changes of $I$ in the freezing process of the nanoparticle are approximately linear right after the initial transition. These quantitative analyses on the phase transitions may provide insight into the control of melting and freezing processes. However, further work is needed to verify and understand these results.

## Discussion

Based on the above analysis of the melting and freezing processes, it is clear that both processes are facilitated by the presence of a sequential intermediate states that intimately link the solid and liquid phases. In this sense, the two opposite processes of melting and freezing are essentially similar. In the reversible phase transitions, superheated/supercooled particles undergo local structure interactions (vacancy formation and aggregation in melting or ordered liquid formation in freezing) that lead to a relaxation of the structure to a similar disordered premelting or prefreezing transition state. Melting and freezing can then

occur collectively in this disordered crystal state, indicating that the final phase transitions are determined by the state of the entire system. The observed trajectories highlight the unique contributions of premelting/prefreezing states, which helps improve the understanding of microscopic mechanism of the phase transitions.

Based on our observations, an interaction–relaxation model is proposed to qualitatively understand the microscopic mechanism that controls the phase transitions. In the formation processes of the constrained states (faceted droplet in Fig. 2k and spherical crystal in Fig. 3f), all types of interactions of different entities of the system make contributions to the increase in conformation stresses. The internal stress reaches a maximum with the formation of faceted droplet or spherical crystal, providing the driving force in the relaxation process of the phase transitions. By relaxation of internal stress, the nanoparticle disengages from the constraint states. The relaxation process regulates the tuning of the nanoparticle between spherical and facetted morphology to complete the corresponding phase transition, indicating that the shape change in phase transitions is affected by bulk transitions. This finding represents an important contribution towards fully understanding the atomic mechanisms of phase transitions and should be included in future theoretical investigations. Although the phase transitions initiated by electron-beam irradiation might be more or less different from those by common approaches, such as thermal heating, the isolating and studying of the intermediate states in both melting and freezing processes by our atomic scale imaging will provide new insight into the fundamental microscopic mechanism of phase transitions.

## Methods

**Sample preparation.** The synthesis of $SrBi_2Ta_2O_9$ has been described in our previous work by a molten salt method[29]. Briefly, $SrBi_2Ta_2O_9$ was synthesized by heating a stoichiometric mixture of $Sr(NO_3)_2$ (99%), $Bi_2O_3$ (99.9%), $Ta_2O_5$ (99.9%), NaCl (99.9%) and KCl (99.9%) in an alumina crucible at 850 °C for 3 h, in which NaCl and KCl (1:1 molar ratio) were used as solvent when melt. The synthesis precursor and the two chlorides were mixed at 1:1 weight ratio. The obtained light yellow particles were washed with deionized water, and then dried at 50 °C for 3 h.

**HRTEM observation and EDS analysis.** HRTEM observations were performed on a JEOL 2,100 electron microscope. The accelerating voltage of the HRTEM was 200 keV. The obtained $SrBi_2Ta_2O_9$ sample was dispersed into alcohol and was drop-cast on a TEM grid coated by carbon film. After drying the grid in air, it was then placed in the electron microscope. The growth and phase transition of Bi nanoparticle was induced by the electron-beam irradiation on the $SrBi_2Ta_2O_9$ platelet. Simultaneously, the melting and freezing trajectories were recorded as videos using a CCD camera (Gatan 832) at atomic resolution. A FEI Tecnai F30

electron microscope was used to record EDS spectrum of the Bi nanoparticle, which is produced under electron-beam irradiation in the TEM for 20 min.

**DSC analysis.** The growth of Bi nanoparticles on $SrBi_2Ta_2O_9$ was achieved by a photoreaction method according to our previous report (see the details in Supplementary Fig. 5)[33]. The melting and freezing temperatures of Bi nanoparticles on the $SrBi_2Ta_2O_9$ substrate were analysed using a DSC instrument (NETZSCH STA 449F3). The sample was heated in an alumina crucible under nitrogen flow from room temperature to 300 °C and then immediately cooled down. The heating and cooling rates are 5 °C min$^{-1}$.

**Data availability.** The data that support the main findings of this study are available from the corresponding authors upon reasonable request.

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

## Acknowledgements

This work was supported by the National Natural Science Foundation of China (Grant Nos 21643012, U1403193 and 21473248), the 'Western Light' Program of Chinese Academy of Sciences (Grant No.YB201303), the Outstanding Young Scientist Program of CAS, and the CAS/SAFEA International Partnership Program for Creative Research Teams. We thank Dr Hengshan Qiu for constructive discussions on the calculation of the crystalline order of the nanoparticle.

## Author contributions

Y.L. and C.W. conceived and designed the research plan, Y.L. carried out the experiments; Y.L., L.Z., D.L.J, J.Z. and C.W. analysed the data. Y.L. wrote the manuscript with help from L.Z., D.L.J, X.Y. and C.W.; Y.L. and C.W. supervised the work, and all authors discussed the results.

## Additional information

**Competing financial interests**: The authors declare no competing financial interests.

**How to cite this article**: Li, Y. *et al. In situ* study on atomic mechanism of melting and freezing of single bismuth nanoparticles. *Nat. Commun.* **8**, 14462 doi: 10.1038/ncomms14462 (2017).

**Publisher's note**: 

