## [Peer Review File · Nature Communications]

Reviewers' comments:

Reviewer #1 (Remarks to the Author):

A. Summary of key results.

The authors use an ingenious experimental setup to observe repeated melting and freezing of Bi nanoparticles (NP) at atomic resolution. They infer that defect-induced melting occurs throughout the bulk, being preceded by an isostructural solid-solid transition, and that freezing is preceded by an intermediate periodically ordered liquid.

B. Originality and Interest.

Being able to directly observe melting and freezing on such fine length and time scales is certainly very exciting, and as far as I know, new. Understanding these phase transitions, both in bulk and in NP's is certainly of broad interest.

C. Data and methodology.

The experimental approach of using the TEM to both heat and image the NP's is really quite clever, especially considering that the heating is tuned to produce a melting and freezing cycle. The resulting images/videos are then presented and interpreted, with not much further refinement apart from Fourier analysis of the image. The results therefore are more qualitative than quantitative.

D. Appropriate use of statistics and treatment of uncertainties.

There are no quantities reported requiring statistical treatment of uncertainties. However, it is not immediately clear that interpretation of TEM images is devoid systematic uncertainty. For example, in Figure 1b the authors point out with blue arrows uniformly grey portions of the NP that are interpreted as portions still in the liquid phase, and use this as evidence for long range orientational correlation between precritical clusters. However, one sees similar grey patches in Fig 1a, where the whole NP is presumably liquid, or even in Fig 1d where it is crystallized.

E. Conclusions: robustness, validity reliability.

I find it very exciting that the authors have found a system and experimental technique that capture crystallization and melting at such fine spatial and temporal scales. From the images presented, the authors present several findings: the above-mentioned long range correlation between precritical clusters, point-defect-induced melting, a "faceted liquid" (Fig. 2f) intermediate state along the melting pathway, and an "ordered liquid" (Fig. 3b) freezing pathway intermediate. The idea that point defects are important in melting dynamics is not new (e.g. Zhang et al J. Chem. Phys. 138, 12A538 (2013)), but it is good to see direct experimental evidence.

However, the other, more exotic findings are not founded on very compelling evidence. The "faceted liquid" and the correlated pre-critical clusters, on the contrary, invoke a rather strong suspicion that the TEM images do not sufficiently characterize the three-dimensional internal structure of the NP, and/or can not fully discern between liquid and crystal. Both the "facted liquid" and the pre-critical cluster correlations are more simply explained by an inability of the TEM to reveal crystallinity, e.g. because of internal NP strucutre (non-monocrystalline order) or because of surface disorder. The "ordered liquid" is also rather strange, and would benefit form deeper analysis with complementary techniques. Might the heat released during crystallization, being unable to escape because of poor thermal conductivity, play a role in the process? That temperature is variable during the phase changes is also problematic for developing a free energy model (Fig 4).

F. Suggested improvements.

The manuscript presents an exciting system, viewed on small time and length scales, that captures

the freezing and melting processes in a recurring cycle. Since the TEM images do not fully characterize what is happening in the NP, the findings discussed above appear to be rather speculative - exciting if true - and provide starting points for future studies to confirm or disprove. To my mind, the manuscript would be more compelling if the authors framed the exotic results (pre-critical cluster correlation, "faceted liquid", "ordered liquid" and their free energy pathway model) as possibilities borne from direct observation, rather than conclusively established.

G. References.

Fine.

H. Clarity and context.

Fine.

Reviewer #2 (Remarks to the Author):

The authors present movies obtained by high res TEM of nanoparticles melting and freezing. The crystalline state is clearly distinguished from the liquids by the presence of lattice fringes, that are towards the limit of what they can resolve but are clearly present. Their temporal resolution is sub-second. These results are interesting, although their approach is not completely novel, for example Nielsen et al. (Science 2014) did similar looking work in a different system, calcium carbonate as opposed to the authors' Bi. The authors analyse the results in a descriptive qualitative way, and frequently refer to classical nucleation theory (CNT).

Although the results are interesting, there are problems with this approach and with the paper:

1) To get their images their samples are subject to a very large flux of electrons (to do the high res TEM), and these electrons are strongly perturbing the system, indeed they can melt the Bi nanocrystals, as the authors say. I appreciate the fact that these electrons are needed to image but the fact that the study is under these extreme conditions greatly reduces the applicability of the work. Typically we are interested in crystallisation in systems unperturbed by large electron fluxes. This is a drawback of this work, but in itself does not rule out publication - in my opinion.

2) The authors invoke CNT frequently. It is not clear to me that CNT, a close-to-equilibrium theory, is reasonable in systems pushed strongly out of equilibrium by a large electron flux. Within CNT the rate is determined by the free energy of a rare nucleus at the top of the barrier. This nucleus could be very sensitive to high electron fluxes. Studying crystallisation in electron beams is a valid thing to do, but applying close-to-equilibrium theories uncritically to analyse the results is probably not a good idea. In CNT the driving force for nucleation is kT , i.e., thermal energy. It is not clear to me that this is a good approximation in systems pushed out of equilibrium by large electron fluxes. It is possible that the electron fluxes can be modelled by a simple increase in temperature, but as far as I know nobody has shown this, and it may not be true.

3) The results are essentially entirely qualitative. This is a pity as they clearly have enough information in their images to approximately quantify crystalline ordering in their system, in a similar way to that done in computer simulation studies, such as those of Frenkel and coworkers (e.g., in their reference 6). If they could quantify the amount of crystalline order as a function of time, they would then have at least semi-quantitative data on crystallisation. This would be an advance on earlier work. It would also allow them to back up their currently very speculative comments (e.g., on the role of what they assume are strains present at certain times) on the pathway from liquid to crystal. At the moment their results do not go much beyond the qualitative statement that Bi nanoparticles can be

melted and then crystallised under a strong electron beam. This is not very surprising. It does not help us understand crystallisation under common lab or industrial conditions, and it does not make predictions as to how to control or improve the crystallisation of Bi nanoparticles.

With order parameters to quantify the crystalline ordering, as a function of time, and with the CNT references either removed or justified, the results could be published, but I do not recommend publication in the current form.

Reviewer #3 (Remarks to the Author):

This manuscript describes an experimental work which captures the kinetic pathways during melting/freezing of bismuth nanoparticles through high resolution TEM technics. The reversible freezing/melting process and their distinct intermediate states are interesting, and may have potential impacts on experimental investigations of phase transition kinetics at atomic scale. However, I have a few comments listed below, which preclude publication of this manuscript at current stage.

1. The authors find same crystal direction in Fig.1b. In Fig.3c, I can still see strong layering feature. The author explains Fig.1 as CNT and Fig.3 as two step nucleation. How to explain the similar layering feature in both figs?
2. The author investigate melting/freezing process of the nano droplet. It is known the nucleation type(homogeneous or heterogeneous) strongly depends on the both the droplet shape and the droplet size. The author should claim the type of the nucleation and the effects of droplet size.
3. During investigation of melting pathways in superheated crystals, the system should keep in constant temperature above melting but below the superheating limit. The author must verify at which temperature region balanced heat flow achieves.

After these questions are satisfactorily addressed, the manuscript can be accepted.

Responses to Reviewers' Comments

First of all, we would like to thank the reviewers for their valuable comments/suggestions on our manuscript. The following are our point-to-point responses.

Reviewer 1

Comment (1):

C. Data and methodology.

The experimental approach of using the TEM to both heat and image the NP's is really quite clever, especially considering that the heating is tuned to produce a melting and freezing cycle. The resulting images/videos are then presented and interpreted, with not much further refinement apart from Fourier analysis of the image. The results therefore are more qualitative than quantitative.

Response:

According to the reviewer's suggestion, we have done our best to quantify the crystalline ordering and the temperature region balanced heat flow in the reversible phase transitions. Please see the details in the following responses.

Comment (2):

D. Appropriate use of statistics and treatment of uncertainties.

There are no quantities reported requiring statistical treatment of uncertainties. However, it is not immediately clear that interpretation of TEM images is devoid systematic uncertainty. For example, in Figure 1b the authors point out with blue arrows uniformly grey portions of the NP that are interpreted as portions still in the liquid phase, and use this as evidence for long range orientational correlation between precritical clusters. However, one sees similar grey patches in Fig 1a, where the whole NP is presumably liquid, or even in Fig 1d where it is crystallized.

Response:

(1) About "quantitative analysis"

Quantitative analysis on the evolution of the order parameter of the individual nanoparticles as a function of time is shown in Fig. 5. Please see the details in the response of comment (3) of the second reviewer.

(2) About "long-range orientational correlation"

We have now revised the wording so as to avoid misunderstanding on the long-range orientational correlation. In fact, the temporally crystallized clusters in the nanodroplet share the same crystal direction as shown in the inset FFT pattern in Fig. 1b, which evidences the long-range ordering interactions between precritical clusters, rather than the presence of the uniformly grey portions in the nanoparticle. To avoid the misleading wording, we changed "precritical clusters" into "**crystallized** clusters" in the revised manuscript (please see the first paragraph of page 7, in revised manuscript).

(3) About "statistics and treatment of uncertainties"

In order to treat uncertainties, sequential snapshots of HRTEM imaging were sampled every second during the metastable state over a period of 17 s and analyzed by FFT. The results are shown in Supplementary Fig. 2. Consequently, the following explanation is added in the revised manuscript (please see the first paragraph of page 7).

To support the long-range orientational correlation in the metastable nanoparticle, sequential snapshots of HRTEM imaging were sampled every second during the metastable state over a period of 17 s (Supplementary Fig. 2). In this period, the nanoparticle was in a continual flux of crystallite formation and dissolution. As seen in the associated FFT patterns, there is a single orientation throughout the entire time signifying long range ordering across the particle consistent across a relatively long time frame.

Comment (3):

E. Conclusions: robustness, validity reliability.

I find it very exciting that the authors have found a system and experimental technique that capture crystallization and melting at such fine spatial and temporal scales. From the images presented, the authors present several findings: the above-mentioned long range correlation between precritical clusters, point-defect-induced melting, a "faceted liquid" (Fig. 2f)

intermediate state along the melting pathway, and an "ordered liquid" (Fig. 3b) freezing pathway intermediate. The idea that point defects are important in melting dynamics is not new (e.g. Zhang et al J. Chem. Phys. 138, 12A538 (2013)), but it is good to see direct experimental evidence.

However, the other, more exotic findings are not founded on very compelling evidence. The "faceted liquid" and the correlated pre-critical clusters, on the contrary, invoke a rather strong suspicion that the TEM images do not sufficiently characterize the three-dimensional internal structure of the NP, and/or can not fully discern between liquid and crystal. Both the "facted liquid" and the pre-critical cluster correlations are more simply explained by an inability of the TEM to reveal crystallinity, e.g. because of internal NP structure (non-monocrystalline order) or because of surface disorder. The "ordered liquid" is also rather strange, and would benefit from deeper analysis with complementary techniques. Might the heat released during crystallization, being unable to escape because of poor thermal conductivity, play a role in the process? That temperature is variable during the phase changes is also problematic for developing a free energy model (Fig 4).

Response:

(1) About the "faceted liquid"

Yes, the TEM imaging cannot characterize the three-dimensional structure of the nanoparticle because of the lacking of the out-of-plane resolution, thereby prohibiting a full description of the internal structure of the nanoparticle. As we know, the electrons in TEM can penetrate the samples with a thickness of about 100 nm (Angew. Chem. Int. Ed. 2011, 50, 2014–2034). For the nanoparticle with a maximum thickness of about 15 nm in the present work, although we cannot fully discern structures between internal nanoparticle and surface by TEM imaging, the crystallized clusters should be easily detected if they indeed exist regardless of if they are internal to or on the surface of the nanoparticle. In some sense, TEM image can also be considered as a 2D projection of 3D features as proven by others in TEM imaging of metal nanoparticles. Two such TEM images of Ag@SiO₂ and Au/Cu₂FeSnS₄ core-shell nanostructures from literature are shown below (Figs 1 and 2), from which the structure of the core could be clearly seen. Moreover, in our manuscript, the observation of

HRTEM lattice fringe images in Fig. 2a clearly proves that the crystallized clusters could be observed under the current experimental conditions. Therefore, uniform contrast of the faceted nanoparticle and the FFT pattern of the HRTEM image in Fig. 2f of our study indicate that the main body of the nanoparticle is with liquid nature at this time. More importantly, the similar phenomenon has been observed on micron-sized Bi. (Pavlovska, A., Dobrev, D., & Bauer, E. *Surf. Sci.* **286**, 176–181 (1993))

According to the reviewer's suggestion, we did not conclusively state the formation of faceted liquid. Instead, we indicate that the main body of faceted nanoparticle in Fig. 2f is liquid. The following sentence was added in the revised manuscript (please see the second paragraph of page 11).

In some sense, a TEM image can be considered as a two-dimensional projection of three-dimensional features. As we know, the electrons in TEM can penetrate the samples with a thickness of about 100 nm³⁵. For the nanoparticle with a maximum thickness of about 15 nm in the present work, although we cannot fully discern structures between internal nanoparticle and surface by TEM imaging, the crystallized clusters should be easily detected if they indeed exist regardless of if they are inside or outside the nanoparticle. Moreover, the observation of HRTEM lattice fringe images in Fig. 2a clearly prove that the crystallized clusters could be observed under the current experimental conditions. Therefore, uniform contrast of the faceted nanoparticle and the FFT pattern of the HRTEM image in Fig. 2f indicate that the main body of the nanoparticle is in a liquid state at this time.

[UNPUBLISHED DATA REMOVED BY EDITORIAL TEAM AS PER AUTHOR REQUEST]

(2) About the "ordered liquid"

Except TEM imaging, X-ray scattering is another technique to detect the structure of ordered liquid. However, based on the state-of-the-art of *in situ* TEM, it is almost impossible to introduce the X-ray scattering techniques to the *in situ* TEM. Up to now, we have not seen any relevant reports. In fact, for the same sample, it is generally accepted that the contrast in the HRTEM image directly reflects the thickness of the sample. Moreover, the gradual formation trajectory of the ordered liquid shown in Figs 3 a-c also provides indirect evidence

for the formation of the ordered liquid. However, we think it will be better to regard the ordered liquid as possibility according to the reviewer's suggestion. Therefore, the following sentence was added in the revised manuscript.

These results to some extent prove the formation of the ordered liquid in Fig. 3c. (please see the end paragraph of page 13)

In order to study the formation mechanisms of the ordered liquid, we have searched some relevant literatures. The thermal capillary waves, or periodic waves caused by heat flux are proposed as the possible reasons for the formation of the ordered liquid. However, based on the present experimental conditions, we cannot determine the mechanisms for the formation of the ordered liquid. Therefore, the following explanation is added in the revised manuscript.

However, the underlying mechanisms for the formation of the ordered liquid remain unclear. Thermal capillary waves or periodic waves caused by heat flux are proposed as the possible reasons involved therein^{38,39}, but these conjectures could not be confirmed. (please see the end of page 13)

(3) About the "free energy model"

According to the reviewer's good suggestion, we emphasized on the landscape of the melting and freezing pathways as possibility. The "proposed energy landscape" was changed into "**possible** energy landscapes" in the revised manuscript (please see the first paragraph of page 15).

Comment (4):

F. Suggested improvements.

The manuscript presents an exciting system, viewed on small time and length scales, that captures the freezing and melting processes in a recurring cycle. Since the TEM images do not fully characterize what is happening in the NP, the findings discussed above appear to be rather speculative-exciting if true-and provide starting points for future studies to confirm or disprove. To my mind, the manuscript would be more compelling if the authors framed the exotic results (pre-critical cluster correlation, "faceted liquid", "ordered liquid" and their free energy pathway model) as possibilities borne from direct observation, rather than conclusively established.

Response:

The pre-critical cluster correlation was determined by analysis of the HRTEM images in a relatively long time of 17 seconds. The faceted liquid, ordered liquid and their free energy pathway model were all reframed according to the reviewer's suggestion. Please see the details in the above point-to-point responses. Furthermore, the phase transitions of the nanoparticle were quantified by calculating the crystalline order of the entire nanoparticle as a function of time. The linear fit of the crystalline order of the entire system over time were found in the two melting and freezing processes, giving us a rate of the phase transition. Please see details in Fig. 5 and the last paragraph in "Discussion section".

Reviewer 2

Comment (1)

To get their images their samples are subject to a very large flux of electrons (to do the high res TEM), and these electrons are strongly perturbing the system, indeed they can melt the Bi nano-crystals, as the authors say. I appreciate the fact that these electrons are needed to image but the fact that the study is under these extreme conditions greatly reduces the applicability of the work. Typically we are interested in crystallization in systems unperturbed by large electron fluxes. This is a drawback of this work, but in itself does not rule out publication - in my opinion.

Response:

The drawback of the *in situ* HRTEM is pointed out in the revised manuscript, which is shown in the following. (please see the end of first paragraph in page 16)

Although the phase transitions initiated by electron-beam irradiation might be more or less different from those by common approaches, such as thermal heating, the isolating and

studying the intermediate states in both melting and freezing processes by our atomic scale imaging will provide new insight into the fundamental understanding on microscopic mechanism of phase transitions.

Comment (2)

The authors invoke CNT frequently. It is not clear to me that CNT, a close-to-equilibrium theory, is reasonable in systems pushed strongly out of equilibrium by a large electron flux. Within CNT the rate is determined by the free energy of a rare nucleus at the top of the barrier. This nucleus could be very sensitive to high electron fluxes. Studying crystallization in electron beams is a valid thing to do, but applying close-to-equilibrium theories uncritically to analyze the results is probably not a good idea. In CNT the driving force for nucleation is kT , i.e., thermal energy. It is not clear to me that this is a good approximation in systems pushed out of equilibrium by large electron fluxes. It is possible that the electron fluxes can be modelled by a simple increase in temperature, but as far as I know nobody has shown this, and it may not be true.

Response:

It is easily to understand that the knock on effect of electron flux can cause the atom displacement in the nanoparticle. However, this atom displacement is irreversible. The reversible phase transitions of Bi nanoparticles observed in the present work indicate that the phase transition might be mainly dominated by temperature change. In fact, there have been many reports on the heating of a sample by electron-beam irradiation (Ref. 19, 28, 31, and 32). Therefore, the interaction between the nanocrystal and the electron beam can be approximately modelled by a simple increase in temperature while imaging by TEM. According to the reviewer's suggestion, we have emphasized this issue in the revised manuscript (please see the end of page 7), which is shown in the following.

Plenty of studies have found that the electron-beam irradiation in TEM can heat samples due to the inelastic scattering of incoming electrons^{19,28,31,32}. While heating with an electron-beam irradiation is physically and mechanistically different than thermal heating, the use of the electron beam as a heater has previously been reported to induce a reversible phase

transitions of Bi nanoparticles²⁸, and a structural transformation of a Cu₂S nanorod¹⁹. Therefore, in the present study, the electron-beam induced phase transitions of Bi nanoparticles while imaging by TEM is proposed to be similar to a simple heating effect.

Comment (3)

The results are essentially entirely qualitative. This is a pity as they clearly have enough information in their images to approximately quantify crystalline ordering in their system, in a similar way to that done in computer simulation studies, such as those of Frenkel and coworkers (e.g., in their reference 6). If they could quantify the amount of crystalline order as a function of time, they would then have at least semi-quantitative data on crystallization. This would be an advance on earlier work. It would also allow them to back up their currently very speculative comments (e.g., on the role of what they assume are strains present at certain times) on the pathway from liquid to crystal. At the moment their results do not go much beyond the qualitative statement that Bi nanoparticles can be melted and then crystallized under a strong electron beam. This is not very surprising. It does not help us understand crystallization under common lab or industrial conditions, and it does not make predictions as to how to control or improve the crystallization of Bi nanoparticles.

Response:

According to the reviewer's suggestion, we quantify the amount of crystalline order as a function of time by FFT analysis. The details are shown in the Supplementary Fig. 9. Based on the FFT analysis, the order parameters of the individual nanoparticles in melting and freezing processes are defined and calculated. Quantitative analysis on the evolution of the order parameter of the individual nanoparticles as a function of time is shown in Fig. 5 in the revised manuscript. The linear fit of the crystalline order of the entire system over time were found in the two melting and freezing processes, giving us a rate of the phase transition. Accordingly, the following is added in the revised manuscript. (please see the end of page 16 and the content in page 17).

In the models shown in Fig. 4, the melting and freezing of the nanoparticle should be regarded as cooperative "multiscale" processes, revealing the importance of collective behavior that is inaccessible in previous experiment. Therefore, finding a suitable parameter

that can characterize the overall feature of the transforming system is a crucial step to estimate the phase transitions. It is known that phase transitions between liquid and solid describes a transformation between a disordered and an ordered state. Therefore, as shown in Fig. 5, the phase transitions of the nanoparticle were quantified by calculating the crystalline order of the entire nanoparticle as a function of time. The amount of crystalline order of the nanoparticle can be determined from the FFT patterns of the two-dimensional TEM images. Commercial software (WSxM 5.0 Develop 6.0) was used for image processing and analysis according to previous reports⁴⁰⁻⁴². The Fourier spectrum is essentially the sum of the FFT outputs from individual atom arrays. The signal quality in the Fourier spectrum was quantified as the peak signal-to-noise ratio R' , which was calculated as the ratio of the peak harmonic intensity to the background noise between 0 and 2 nm⁻¹ (see details in Supplementary Fig. 9). To quantify crystalline order, we derived the order parameter equation of the nanoparticle, which is given by $S = (R'-1)/(R-1)$ where R is the signal-to-noise ratio of a defect-free nanocrystal. Nanoparticles having a perfect lattice would have $S = 1$, whereas liquid nanodroplet would have S approaching 0. In the present study, the nanoparticle in Fig. 2a is defined as a perfect crystal ($R = 54.4$) with $S = 1$. As shown in Figs 2f-i, the FFT spots are indistinguishable from the background, and the S values for these images are defined as 0. In the reversible phase transitions, the crystalline order for a nanoparticle is quantified as a function of time by calculating the parameter S (Fig. 5), indicating that the changes in S are approximately linear right after the initial transitions. We use this to define the phase transition rate r_t of the nanoparticle (during linear fit) as $r_t = \Delta S / \Delta t$. The transformation rates in melting and freezing of the nanoparticle are ~ 0.46 s⁻¹ and ~ 0.73 s⁻¹, respectively. The linear fit of the crystalline order of the entire system over time may provide insight into the control of melting and freezing processes. However, further work is needed to verify and understand these results.

Comment (4)

With order parameters to quantify the crystalline ordering, as a function of time, and with the CNT references either removed or justified, the results could be published, but I do not recommend publication in the current form.

Response:

According to the reviewer's suggestion, crystalline ordering has been quantified (Please see the details in the response of question 3), and the CNT references is justified (Please see the details in the response of question 2).

Reviewer 3

Comment (1)

The authors find same crystal direction in Fig.1b. In Fig.3c, I can still see strong layering feature. The author explains Fig.1 as CNT and Fig.3 as two step nucleation. How to explain the similar layering feature in both figs?

Response:

In Fig.1, the liquid Bi nanodroplet actually exists in a metastable liquid state in which frequent formation and dissolution of crystallized clusters were observed inside the nanodroplet. These layering feature in Fig.1 is only a transient phenomenon, and not ordered, which did not play a direct effect on the crystallization process. However, layering structure in Fig.3 is relatively stable with highly ordered, which plays an important role in the crystallization process. These differences between the two structures have been described in the manuscript.

In fact, the phenomenon in Fig. 3 was not referred as a two-step nucleation process, but as a two-step crystallization process by an intermediate ordered liquid. The nucleation towards

the crystallization of the nanodroplet is initiated in the ordered liquid. As indicated in the manuscript, the formation of ordered liquid at this stage allows for a large driving force (i.e., free energy change) to form a prefreezing state with disordered structure. Subsequently, the crystallized layers in the prefreezing state interact with each other to develop a collective behavior of nucleation and growth of the nanocrystal from the outer surface (Figs 3d-g), which results in the gradual formation of a well faceted nanocrystal. The formation of ordered liquid (Fig. 3c) and prefreezing state (Fig. 3d) were regarded as the intermediate states in the prenucleation stage.

We inferred that the observation in Fig. 1 is similar to the classical nucleation model because crystalline embryos continuously fluctuate their sizes and shape before reaching a critical nucleus size. Therefore, both of the formation processes of the crystallized clusters in Figs 1a-b and the layering structure with disordered structure act as prenucleation stages of the crystallization process. In this sense, both of the two crystallization processes are thermodynamically similar. However, the kinetics of these two crystallization processes is different because of the different thermal contact areas between the Bi liquid and the support before and after the first crystallization of the nanoparticle at 1099.7 s (see Supplementary Fig. 4).

Comment (2)

The author investigate melting/freezing process of the nano droplet. It is known the nucleation type (homogeneous or heterogeneous) strongly depends on the both the droplet shape and the droplet size. The author should claim the type of the nucleation and the effects of droplet size.

Response:

(1) About the nucleation type

According to the reviewer's suggestion, the nucleation types in melting and freezing process of the nanoparticle was indicated. The nucleation in melting of the nanoparticle (Fig.2) is heterogeneous, which is indicated in the first paragraph of page 11 in the revised manuscript. The nucleation in freezing of the nanoparticle (Fig. 3) is homogeneous, which is

indicated in page 14 in the revised manuscript.

(2) About the effects of droplet size on nucleation type

It is true that the nucleation type is influenced by many factors, such as droplet size. However, we focus on studying the melting and freezing mechanism of the same bismuth nanoparticle in this work. The effect of particle size on the nucleation type is not the topic discussed in this article. Therefore, according to our current understanding, it might not be necessary to claim the effects of droplet size on the nucleation. This interesting issue should be the subject of our further investigations.

Comment (3)

During investigation of melting pathways in superheated crystals, the system should keep in constant temperature above melting but below the superheating limit. The author must verify at which temperature region balanced heat flow achieves.

Response:

In order to verify the temperature region balanced heat flow in the reversible phase transitions, we prepared the Bi nanoparticles on the $\text{SrBi}_2\text{Ta}_2\text{O}_9$ substrate by a photoreduction method. The temperature region is roughly determined to be 160-188 °C by differential scanning calorimetry measurements. The DSC results are added as Supplementary Fig. 6. The following content was added in the revised manuscript (please see the second paragraph of page 9).

To understand the temperature and heat flow properties of this system, the temperature range of melting and heating is analyzed. The melting temperature of the Bi nanoparticles is known to be proportional to the reciprocal of the radius of the particles²², and the radius of the nanoparticle shown in Fig. 1 is determined to be about 8 nm. Differential scanning calorimetry (DSC) was used to determine the temperature range over which this reversible reaction occurs. However in order to achieve the large density of nanoparticles needed for a DSC signal, the Bi particles were grown on the $\text{SrBi}_2\text{Ta}_2\text{O}_9$ substrate via a photoreduction method introduced in our previous work³³. The TEM image (Supplementary Fig. 5) shows that the radius of the nanoparticles formed on the surface of the platelet after 40 h

photoreaction is in the range of 6-9 nm, which is similar with that in Fig. 1. The heating and cooling DSC curves (Supplementary Fig. 6) indicate that the onset melting and onset freezing temperature of these nanocrystals are 188 °C and 160 °C, respectively. The particles grown in the photoreduction method exhibit a small size variation spanning across the particle size grown under electron beam radiation. Thus, the phase transition temperature range produced under electron beam irradiation on the single particle formed under HRTEM should be reasonably close to the measured range of 160 °C-188 °C, assuming the effect of the electron beam can be modeled as a simple heating source as argued above.

Accordingly, the experimental method for the DSC analysis is added in the revised manuscript (please see the end of page 18).

DSC analysis. The growth of Bi nanoparticles on SrBi₂Ta₂O₉ was achieved by a photoreaction method according to our previous report (see the details in Supplementary Fig. 5)³³. The melting and freezing temperatures of Bi nanoparticles on the SrBi₂Ta₂O₉ substrate were analyzed using a DSC instrument (NETZSCH STA 449F3). The sample was heated in an alumina crucible under nitrogen flow from room temperature to 300 °C and then immediately cooled down. The heating and cooling rates are 5 °C min⁻¹.

Reviewers' comments:

Reviewer #1 (Remarks to the Author):

My concerns with the original manuscript were over some aspects of the rather qualitative analysis of the TEM images, uncertainty over the ability of TEM to sufficiently characterize the nanodroplets, and too much emphasis on some less firmly grounded components of the work.

The authors have addressed all of these points, so I now recommend publication.

They have added estimates of transformation rates from time series of crystallization order parameters, added discussion and references on the capabilities of TEM, and provided more Fourier analyses of TEM images. They have also provided discussion and references with regard to possible mechanisms to the "ordered liquid", while more carefully describing this and other elements of their work (free energy description, faceted liquid, pre-critical cluster correlation).

Reviewer #2 (Remarks to the Author):

I would like to thank the authors for their reply to my comments. The method they now use for quantifying ordering is interesting, and a good development. However, I am not convinced that there are distinct multiple steps in the processes they study, they have no evidence that classical nucleation theory (CNT) is appropriate for their system, and Figure 4 is pure speculation. Also, melting and freezing nanoparticles with the aid of large electron fluxes is interesting, but as this is not how they usually melt or freeze, I do not think it is clearly of wide interest. So I am not able to recommend publication.

A couple of more detailed comments:

1) In my first review, I said that they do not have evidence the CNT - a theory for nucleation via a rare thermal equilibrium fluctuation - is applicable to a system under a huge flux of electrons. The high e flux presumably means the system is far from thermal equilibrium. In their reply they say:

"While heating with an electron-beam irradiation is physically and mechanistically different than thermal heating, the use of the electron beam as a heater has previously been reported to induce a reversible phase transitions of Bi nanoparticles, and a structural transformation of a Cu₂S nanorod. Therefore, in the present study, the electron-beam induced phase transitions of Bi nanoparticles while imaging by TEM is proposed to be similar to a simple heating effect."

This is essentially my point, they have 'proposed' that it is 'similar' to heating, they have not shown it. And the fact that you can use an electron flux to reversibly melt solids does not prove that the mechanism of melting is well described by CNT.

For example, even if you assume that the absorption of large amounts of energy by collisions with electrons does not make the distribution of states very far from the Boltzmann distribution at equilibrium (a big assumption), it is not clear to me that the effective temperature (assuming we can define such a thing) is uniform. For example, if the Bi absorbs more energy than the surroundings then the nanoparticle centre will tend to be hotter than its edge. This presumably will effect the kinetics.

2) Figure 4 is highly speculative. I do not really see evidence for two steps, as I think the intermediate

minima in their free energies imply. Also, as I noted in point 1), there is no hard evidence that CNT applies here. Finally, they talk about 'strain energy'. Strain is a free energy not an energy, if a crystal is deformed, then in general the crystal's entropy and its energy will both change. For example, if you deform a crystal of hard spheres, the energy change will be zero, but there is still a strain free energy due to entropy changes.

3) The novel order parameter in Figure 5 is interesting. I think it comes from the Fourier transform (from real space to k) of a one-body density - if you assume the darker a pixel then the more Bi there is at that point. They take a 1D linescan and Fourier transform what I think is effectively a function proportional to $\rho(z)$ (convolved with some function that describes the imaging resolution), for $\rho(z)$ the Bi density at a point z along the linescan.

This should measure order, but it would not be much more work to measure density-density correlations, i.e., along a linescan, something like $\rho(z, z+\Delta)/[\rho(z)\rho(z+\Delta)]$ as a function of Δ (or perhaps even better the 2D analog). This density-density correlation is what X-ray diffraction measures by the structure factor $S(k)$. Thus if they computed that, they can plot something roughly equivalent to what X-rays would see, and also they would be able to employ the range of well-known methods of estimating ordering from $S(k)$ s. For example, then the peak width (as measured by FWHM) should roughly scale with the size of the ordered domain. This is described by what is called the Scherrer equation, an equation used in XRD analysis.

Reviewer #3 (Remarks to the Author):

The authors have successfully answered all my questions and I agree to publish this manuscript in Nature Communications.

Responses to Reviewers' Comments

Reviewer 2

Comment (1):

In my first review, I said that they do not have evidence the CNT - a theory for nucleation via a rare thermal equilibrium fluctuation - is applicable to a system under a huge flux of electrons. The high e flux presumably means the system is far from thermal equilibrium. In their reply they say:

"While heating with an electron-beam irradiation is physically and mechanistically different than thermal heating, the use of the electron beam as a heater has previously been reported to induce a reversible phase transitions of Bi nanoparticles, and a structural transformation of a Cu₂S nanorod. Therefore, in the present study, the electron-beam induced phase transitions of Bi nanoparticles while imaging by TEM is proposed to be similar to a simple heating effect."

This is essentially my point, they have 'proposed' that it is 'similar' to heating, they have not shown it. And the fact that you can use an electron flux to reversibly melt solids does not prove that the mechanism of melting is well described by CNT.

For example, even if you assume that the absorption of large amounts of energy by collisions with electrons does not make the distribution of states very far from the Boltzmann distribution at equilibrium (a big assumption), it is not clear to me that the effective temperature (assuming we can define such a thing) is uniform. For example, if the Bi absorbs more energy than the surroundings then the nanoparticle centre will tend to be hotter than its edge. This presumably will affect the kinetics.

Response:

We agree with the reviewer that the reversible phase transitions induced by the electron flux may be different from those by thermal heating. It is not clear that CNT is reasonable in the present system. Therefore, the CNT references are removed in the "Results" and "Discussion" parts of the manuscript according to the reviewer's suggestion. The changes are listed below.

Numbers	The sentences in our previous manuscript	The corresponding sentences in the revised manuscript
1	"During the melting and freezing, multiple barrier-crossing events including formation of nucleation precursors, nucleation and growth, and the relaxation of system, are directly observed ascribing to the existence of multiple intermediate states."(in the abstract)	"During the melting and freezing, the formation of nucleation precursors, nucleation and growth, and the relaxation of system, are directly observed."(in the abstract)
2	This observation resembles the classical nucleation model, where crystalline embryos fluctuate their sizes before reaching a critical nucleus size. (in the beginning of page 7)	This sentence has been deleted in the revised manuscript. (in the first paragraph of page 7)
3	Interestingly, the inset FFT pattern in Fig. 1b shows that these crystallized clusters share the same crystal direction, implying that the system has long-range ordering interactions, which is not predicted by the CNT. (in line 6-8 of the first paragraph in page 7)	Interestingly, the inset FFT pattern in Fig. 1b shows that these crystallized clusters share the same crystal direction, implying that the system has long-range ordering interactions. (in the first paragraph of page 7)
4	"The total free energy of the system reaches a maximum at a critical point, at which the defects start to aggregate into a gap (indicated by dashed blue lines) within the nanocrystal to lower the free energy (Fig. 2d)." (in the beginning of page 11)	"Following that, the defects start to aggregate into a gap (indicated by dashed blue lines) within the nanocrystal to lower the free energy (Fig. 2d)." (in the beginning of page 11)
5	" At this time, the total free energy of the system reached a maximum, which induced a subsequent rapid melting of the entire nanoparticle at 1117.5 s (Fig. 2f), as indicated by the FFT pattern." (in the end of the first paragraph in page 11)	"As indicated by the FFT pattern, a subsequent rapid melting of the entire nanoparticle occurs at 1117.5 s (Fig. 2f)" (in the end of the first paragraph in page 11)
6	"The classical nucleation theory of melting process suggests that melting starts at the surface of the solid." (in the beginning of the third paragraph in page 12)	"It is well known that melting should start at the surface of the solid." (in page 12)
7	"The free energy barrier for the formation of a liquid nucleus within the solid phase can be described as equation (1) ⁷ $\Delta G = -V\rho\Delta\mu + A\gamma + E_{\text{strain}} - E_{\text{defect}} \quad (1)$ where V , ρ , A , and γ are the volume, density, surface area, and surface tension of the liquid nucleus, respectively, $\Delta\mu$ is the difference in chemical potential between the solid and liquid phase, E_{strain} represents the strain energy in the solid induced by the volume variations in the liquid nucleus	This paragraph has been deleted in the revised manuscript.

	formation, and E_{defect} is the defect energy pre-existed in volume V . The presence of disorder lowers E_{strain} , while the presence of the defect increases E_{defect} in equation (1) ⁵ . Therefore, the formation of the intermediate premelting state in Fig. 2c reduces the nucleation barrier in the melting process." (in page 12)	
8	"Therefore, the formation of ordered liquid at this stage allows for a large driving force (i.e., free energy change) to crystallization as shown in Fig. 3d, which represents the lowest energy state via enthalpy change." (in line 5-8 of the second paragraph in page 14)	"Therefore, the formation of ordered liquid at this stage allows for a large driving force (i.e., free energy change) to crystallization as shown in Fig. 3d." (in the middle of the second paragraph in page 13)
9	"This prefreezing state reduces the energy barrier for the homogeneous nucleation in the following crystallization process."(in line 12-13 of the second paragraph in page 14)	This sentence has been deleted in the revised manuscript. (in the end of the second paragraph in page 13)

Comment (2):

Figure 4 is highly speculative. I do not really see evidence for two steps, as I think the intermediate minima in their free energies imply. Also, as I noted in point 1), there is no hard evidence that CNT applies here. Finally, they talk about 'strain energy'. Strain is a free energy not an energy, if a crystal is deformed, then in general the crystal's entropy and its energy will both change. For example, if you deform a crystal of hard spheres, the energy change will be zero, but there is still a strain free energy due to entropy changes.

Response:

The descriptions of the nature of the multi-step processes on the phase transitions are disclaimed in the revised manuscript. Accordingly, Figure 4 in our previous manuscript is deleted in the revised manuscript. Furthermore, instead of the "strain energy" referred in our previous manuscript, the internal stress is proposed to be the driving force for the relaxation process of the phase transitions. Please see the details in the "Discussion" part of the revised manuscript.

Comment (3):

The novel order parameter in Figure 5 is interesting. I think it comes from the Fourier transform (from real space to k) of a one-body density - if you assume the darker a pixel then the more Bi there is at that point. They take a 1D linescan and Fourier transform what I think is effectively a function proportional to $\rho(z)$ (convolved with some function that describes the imaging resolution), for $\rho(z)$ the Bi density at a point z along the linescan.

This should measure order, but it would not be much more work to measure density-density correlations, i.e., along a linescan, something like $\rho(z, z+\delta)/[\rho(z)\rho(z+\delta)]$ as a function of δ (or perhaps even better the 2D analog). This density-density correlation is what X-ray diffraction measures by the structure factor $S(k)$. Thus if they computed that, they can plot something roughly equivalent to what X-rays would see, and also they would be able to employ the range of well-known methods of estimating ordering from $S(k)$ s. For example, then the peak width (as measured by FWHM) should roughly scale with the size of the ordered domain. This is described by what is called the Scherrer equation, an equation used in XRD analysis.

Response:

We appreciate the valuable suggestion, which allows us to precisely evaluate the crystalline order of the entire nanoparticle in the phase transitions. According to the suggestion, two-dimensional FFT patterns were integrated into 1D data using the Fit2D program, which is usually used for integrating two-dimensional X-ray diffraction images (see details in Supplementary Fig. S9). The crystalline order for a nanoparticle is quantified as a function of time by calculating the peak intensity of the Fourier spectrum. The detailed discussions on the crystalline order are provided in the end of page 14 and the first paragraph of page 15 in the revised manuscript.

The reviewer's suggestion on further processing the intensity profile of the 2D Fourier pattern by a method that is similar with processing X-ray diffraction data is constructive and meaningful. However, based on the experimental data from the HRTEM images and Fourier analyses in the revised manuscript, the melting and freezing processes would be well understood. Therefore, further processing of the two-dimensional FFT patterns was not carried out. Detailed explanation on this issue is as follows.

As we know, the phase transitions are characterized by the state of the entire system. From the HRTEM images in Fig. 2 and Fig. 3, the melting and freezing of the nanoparticle should be regarded as a collective behavior of the nanoparticle. Therefore, the parameter that can quantify the ordering of the entire nanoparticle is sufficient for characterizing the phase transitions. The employed 2D Fourier analyses in Fig. S9 reveal the patterns that allow the quantitative identification of the ordering of the entire system. Therefore, it might not be necessary to claim the local information of the HRTEM images, such as the density-density correlations of the ordering, in the phase transitions of the nanoparticle.

In the melting and freezing processes, quantifying the overall ordering of the system is not needed to confirm the shape, size, and the distribution of the ordered domains. In case that the averaged grain sizes can be obtained by analyzing Fourier patterns, this method is not straightforward and may be hampered by the irregular shape and wide size distribution (from sub-nanometer to ~10 nm) of the entities in the corresponding HRTEM images in the manuscript. Some information, such as the shape, size and number of the ordered domains can be directly extracted from the HRTEM images because of the very limited number of the ordered domains (less than 5) and the small size of the evaluated nanoparticle (with a radius of 8 nm) in the manuscript. By processing the Fourier spectra, even if the width of the FWHM (full-width at half-maxima) could yield an averaged size of the ordered domain, it might also be regarded as a theoretical value with little use because of the few specimens (less than 5). Therefore, it might be not necessary to further process the Fourier spectra in Fig. S9 to get the average size of the ordered domains in the phase transitions.

The averages in the X-ray peaks are volume averages, whereas the FFT patterns in the present study are transformed by two-dimensional images. Therefore, the methods for analyzing the FFT patterns might be more or less different from those by X-ray data. Except the size effect, the width of the peaks in X-ray studies can also be influenced by many other factors such as instrumental factors (see Ref. 1). Moreover, the factors that affect the Fourier spectra are still not clear because there is little research in this area. Therefore, we are not sure on the suitability of processing the intensity profile of the 2D Fourier pattern by the method based on processing X-ray diffraction data. According to the previous report (see Ref.

2), Fourier analysis is not a suitable means of extracting quantitative information of the local areas of the corresponding HRTEM images as the spatial resolution is lost in the single Fourier component.

Based on the above considerations, further processing of the two-dimensional FFT patterns was not carried out. This interesting issue should be the subject of our further investigations.

References

1. Cohen J.B. X-ray diffraction studies of catalysts. *Ultramicroscopy* **34**, 41-46 (1990)
2. Ourmazd, A., Taylor, D. W., Bode, M. & Kim Y. Quantifying the Information Content of Lattice Images. *Science* **1246**, 1571–1577 (1989)

REVIEWERS' COMMENTS:

Reviewer #1 (Remarks to the Author):

In regard to the response to referee #2, I find that the authors have strengthened the manuscript by removing the speculative links to CNT and multistep nucleation. This experimental work provides enough fertile ground for theorists as it is.

The rewritten portions of the discussion point to sequential intermediates, collective motions, and internal stresses... all of which are interesting.

Therefore, I continue to recommend publication.